# Graph neural networks learn emergent tissue properties from spatial molecular profiles

Mayar Ali[1,2,3,8], Sabrina Richter[1,4,8], Ali Ertürk [2,5], David S. Fischer[1,6,7] ✉ & Fabian J. Theis [1,4,7] ✉

Tissue phenotypes, such as metabolic states, inflammation, and tumor properties, emerge from both molecular states and spatial cell organization. Spatial molecular assays provide an unbiased view of tissue architecture, enabling phenotype prediction. Graph neural networks (GNNs) offer a natural framework for analyzing spatial proteomics by integrating expression profiles with structure. We apply GNNs to classify tissue phenotypes using spatial cell patterns. We show that for relatively simple classification tasks, such as tumor grading in breast cancer, incorporating spatial context does not significantly improve predictive performance over models trained on single-cell or pseudobulk representations. However, GNNs capture meaningful spatial features, retaining prognostic signals beyond tumor labels, highlighting tumor-grade-specific cell type interactions, and uncovering complex immune infiltration patterns in colorectal cancer not detectable with traditional approaches. These findings suggest that while spatial dependencies may not always enhance classification performance in small datasets, GNNs remain valuable tools for characterizing tissue organization and interactions.

The high molecular resolution provided by single-cell RNA-seq (scRNA-seq) has put the cell as a functional unit in the focus of recent advances in tissue biology[1]. However, interactions between cells and properties of the tissue beyond the length scale of a cell are largely lost in assays that are based on dissociated tissues. Highly multiplexed imaging technologies such as Imaging mass cytometry (IMC)[2] and co-detection by indexing (CODEX)[3] enable the simultaneous measurement of dozens of protein markers at subcellular resolution within intact tissues. These technologies are particularly valuable in oncology and immunology, where they help characterize the tumor microenvironment and study how spatial organization of cells shapes disease progression and therapeutic responses[4,5]. By capturing the coordinated behavior of malignant, immune, and stromal cells in different tumor phenotypes, these datasets provide insights into mechanisms of

effective versus ineffective tumor control, ultimately advancing immunotherapies[6,7].

In order to analyze tissue organization, single-cell spatial omics data can be modeled as spatial graphs, where nodes represent individual cells and edges encode spatial proximity. This representation enables computational models to capture tissue architecture and cellular interactions explicitly[8–10]. Graph neural networks (GNNs) have emerged as a powerful tool for integrating spatial, molecular, and cellular information. Recent studies suggest that GNNs can identify disease-relevant tissue structures and even outperform traditional clinical metrics in certain prognostic tasks[11–13]. However, the extent to which GNNs effectively leverage spatial context for prediction, and whether their learned representations faithfully capture biologically meaningful features, remains unclear.

[1]Institute of Computational Biology, Helmholtz Zentrum München, Neuherberg, Germany. [2]Institute for Tissue Engineering and Regenerative Medicine, Helmholtz Zentrum München, Neuherberg, Germany. [3]Graduate School of Systemic Neurosciences, Ludwig Maximilian University of Munich, Munich, Germany. [4]TUM School of Life Sciences Weihenstephan, Technical University of Munich, Freising, Germany. [5]Institute for Stroke and Dementia Research, Klinikum der Universität München, Ludwig-Maximilians-Universität LMU, Munich, Germany. [6]Eric and Wendy Schmidt Center at the Broad Institute, Cambridge, MA, USA. [7]Department of Mathematics, Technical University of Munich, Garching bei München, Germany. [8]These authors contributed equally: Mayar Ali, Sabrina Richter. ✉e-mail: david.fischer@meduniwien.ac.at; fabian.theis@helmholtz-munich.de

Here, we systematically evaluate the predictive performance and interpretability of GNNs for tumor phenotype classification using spatial omics data. We first conduct a comparative multi-model ablation study to assess the individual and combined contributions of spatial context and single-cell features to predictive performance. Second, we perform in-depth interpretability analyses of graph models to understand the underlying factors driving model predictions and to better understand the biological relevance of the learned representations. Specifically, we address two key questions: (1) Does spatial context enhance predictive performance compared to single-cell or bulk representations? (2) Can graph models yield biologically meaningful insights into tissue organization? To this end, we explore several model interpretation strategies, including learned sample embeddings, attention-based interaction patterns, and saliency maps, to determine whether GNNs capture relevant biological structures. Our findings aim to clarify the role of spatial information in tumor phenotype prediction and highlight the potential of graph-based models as interpretable tools for spatial omics and tissue biology.

## Results

### Graph neural networks model tissue phenotypes

To investigate the role of tissue architecture in tumor phenotype prediction, we perform a multi-model ablation study to assess the performance of graph neural networks (GNNs) across multiple spatial omics datasets. Our goal is to determine how spatial context and single-cell information contribute to predictive accuracy, and whether graph-based models can capture meaningful biological patterns across different cancer types and imaging platforms (Fig. 1a, b).

We first evaluate the performance of GNNs in predicting tumor phenotypes from spatial omics data, specifically examining the influence of spatial context and single-cell resolution. For this, we consider three distinct datasets with graph-level supervision tasks: a cohort of CODEX samples from colorectal cancer biopsies (*CODEX - colorectal cancer*[14], 140 images from 35 patients), and two cohorts of imaging mass cytometry (IMC) breast cancer biopsy data (*IMC - Jackson*[15], 559 images from 350 patients and *IMC - METABRIC*[16], 500 images from 454 patients). For the *CODEX - colorectal cancer* dataset, we focus on predicting binary anatomic labels, specifically the presence of tertiary lymphoid structures. For the *IMC - Jackson* and *IMC - METABRIC* datasets, we predict tumor grades, distinguishing between grades 1, 2, and 3 tumors. In all cohorts, hold-out splits are defined by patients to avoid leakage of batch information (Methods).

We represent the data as spatial graphs, where each node corresponds to an individual cell and is annotated with single-cell features. Spatial graphs are constructed by connecting cells with an edge if their Euclidean distance fell below a fixed threshold radius, with

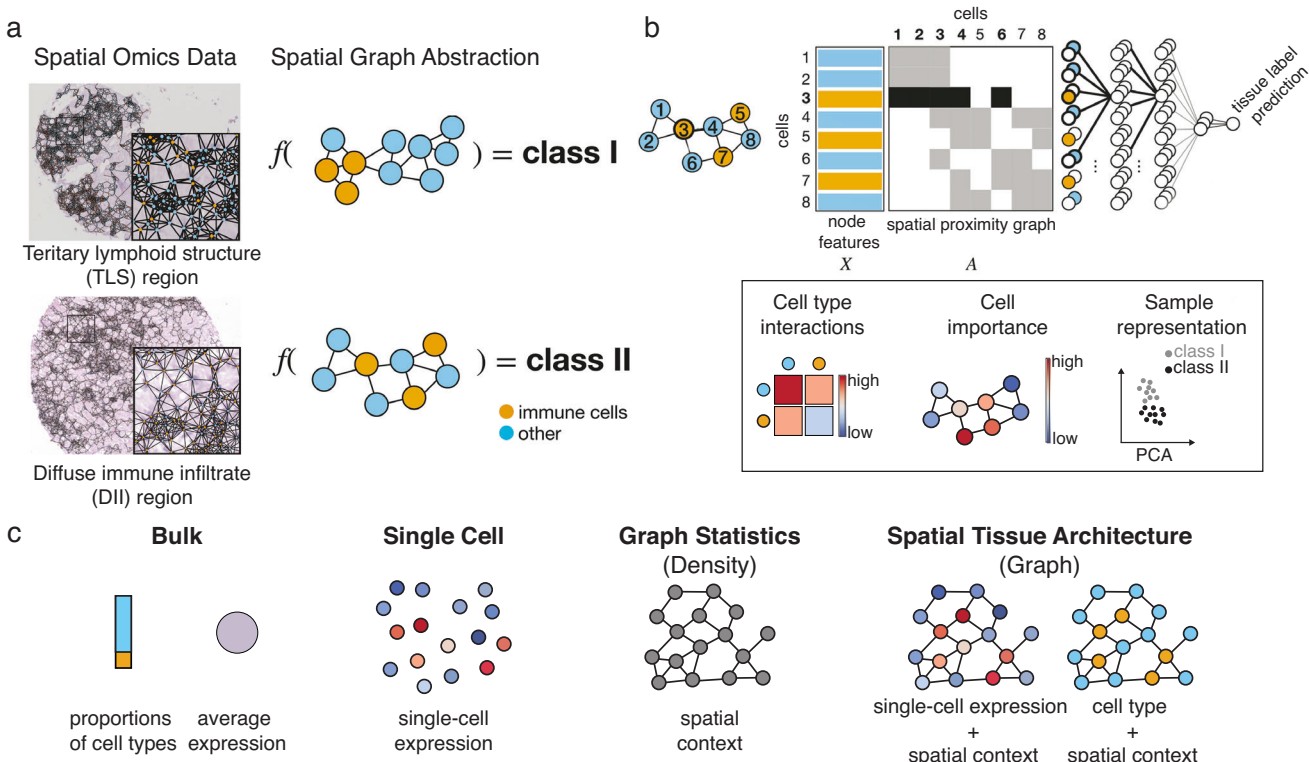

**Fig. 1 | Overview of spatial graph framework and ablation design for phenotype prediction. a** Tissue-level phenotypes are functions of the architecture of the tissue. In this case, two immune infiltration regions, Tertiary lymphoid structure (TLS) region and diffuse immune infiltrate (DII) region, can be distinguished based on the spatial distribution of immune cells. This anatomical label cannot be inferred based on frequencies of cell types that would be available in dissociation-based protocols, but only based on the spatial distribution of cells[14]. One example image from the *CODEX - colorectal cancer* dataset for each class. **b** (top) The spatial context of each cell can be formally represented by a graph in which edges are weighted based on the distance between nodes. Each sample can be represented as one such graph, where nodes are colored by the measured cell features. Node features and the proximity graph are input to the model. We perform prediction with a model that consists of graph neural network layers to produce node embeddings, followed by pooling over nodes and a final classification network that outputs a tissue-level label. (bottom) Different downstream tasks and interpretation approaches can be performed using the graph embeddings, such as cell-type interactions or neighborhood analysis, cell importance to the phenotype prediction, and sample representation where the spatially-aware graph embeddings can be visualized with a PCA in which each point reflects one graph (image) and depicts separation of samples by the tissue-level class. **c** Design of the ablation study. *Bulk* models only have access to the average node feature vector of the graph whether proportions of cell types or average molecular expression. *Single Cell* models have access to single-cell-resolved but in silico dissociated data from the observed spatial graph. *Graph Statistics* are spatially aware models that have access to the full spatially resolved data but reduce it to simpler summary statistics such as tissue density represented by node degree. Finally, *Spatial Tissue Architecture* which are represented by graph models, have access to node features and the spatial proximity graph.

**Table 1 | Hyperparameters related to training and data processing screened in grid search for each dataset**

| dataset | learning rate | l2 | radius | number of clusters |
|---|---|---|---|---|
| IMC - breast cancer (Jackson) | {5e-2, 5e-3, 5e-4} | {0, 1e-6, 1e-3} | {10, 20, 50} | {5, 10, 20} |
| IMC - breast cancer (METABRIC) | {5e-2, 5e-3, 5e-4} | {0, 1e-6, 1e-3} | {10, 20, 55} | {5, 10} |
| CODEX - colorectal cancer | {5e-2, 5e-3, 5e-4} | {0, 1e-6, 1e-3} | {25, 50, 120} | {5, 10} |

neighborhood sizes (resolutions) determined based on the average node degree distribution[3] (Supp. Figure 1). This representation enables the modeling of both cellular attributes and spatial relationships within the tissue. GNNs operate on these graphs by iteratively aggregating information from neighboring nodes and ultimately pooling the learned cell-level representations into a single graph-level embedding, which serves as the basis for tissue phenotype prediction.

## Ablating over tissue architecture motives in spatial omics for tumor phenotype prediction

To disentangle the contribution of spatial context and single-cell resolution to model performance, we designed a comprehensive ablation study building on the spatial graph framework described above. In this setting, we constructed spatial graphs where each node represents an individual cell and is annotated with its molecular profile, such as protein or gene expression levels, while edges represent spatial proximity as previously described. To benchmark the utility of spatial context and single-cell resolution, we compared three scenarios: (1) the full molecular profiles of cells within their spatial context encoded via spatial proximity graphs (Spatial Tissue Architecture), (2) molecular profiles of in silico dissociated single cells without any encoding of potential interactions (Single Cell), and (3) pseudobulk profiles computed as the mean molecular expression across all cells in a tissue image (Bulk) (Fig. 1c). For each of these inputs, we applied tailored machine learning models: graph convolutional networks (GCN) and graph isomorphism networks (GIN) for the spatial graphs, multi-instance learning (MI) models for the single-cell input, and multi-layer perceptrons (MLPs), logistic regression and random forests (RF) for the pseudobulk-level representation. We optimized all models using hyperparameter grid searches within a nested cross-validation framework (Table 1, Table 2), and performance was evaluated using the area under the precision-recall curve (AUPR) to account for class imbalances (Methods).

We found that GNNs trained on spatial graph representations of tissue images did not significantly outperform multi-instance learning (MI) models trained solely on single-cell expression vectors for all datasets (ΔAUPR = 0.052 and $p = 0.21$, ΔAUPR = 0.036 and $p = 0.086$, ΔAUPR = 0.014 and $p = 0.56$ for *CODEX - colorectal cancer, IMC - Jackson* and *IMC - METABRIC*, respectively, Fig. 2a, b, f). In addition, the single-cell resolution modelled by the MI models offered no substantial improvement over pseudobulk representations (ΔAUPR = −0.012 and $p = 0.71$, ΔAUPR = 0.005 and $p = 0.80$, ΔAUPR = −0.021 and $p = 0.31$ for *CODEX - colorectal cancer, IMC - Jackson* and *IMC - METABRIC*, respectively, Fig. 2a, b, f, Supp. Figure 2). Notably, only in the *IMC - Jackson* breast cancer dataset did the spatial model significantly outperform the pseudobulk-level representation (ΔAUPR = 0.041 and $p = 0.019$). These findings suggest that the added value of spatial context or single-cell resolution for tumor phenotype prediction is limited in current spatial omics datasets, which comprise only up to a few hundred images. The strong performance of pseudobulk representations likely reflects their ability to smooth out cell-to-cell variability and emphasize dominant molecular signals at the tissue level. In contrast, more complex spatial or single-cell models may require larger datasets, or more complex phenotypes that are tightly coupled to spatial organization, to fully leverage the additional layers of information they encode.

## Spatially-aware graph embeddings reveal clinically meaningful tissue representations

The ability of graph neural networks (GNNs) to explicitly model cellular interactions and tissue architecture offers unique opportunities for capturing biologically meaningful spatial features that may not directly translate into classification performance. Despite the comparable predictive performances of GNNs to the baseline models, it remains worthwhile to explore what these models learn about tissue organization and whether their learned representations reflect relevant biological structures or processes. Therefore, we analyzed the graph-level embeddings learned by the GNNs. These embeddings, obtained after node-pooling, provide spatially-aware representations of entire tissue samples and can be interpreted as a continuous patient manifold (Methods). Interestingly, the embeddings revealed biologically meaningful patterns beyond the separation required for tumor phenotype classification. For the two breast cancer datasets, we found that the graph embeddings recapitulated the sequential ordering of tumor grades (1, 2, and 3), even though the categorical multi-class loss function does not enforce such ordering. For the IMC - Jackson dataset, the embeddings showed a clear gradient of tumor grades, progressing from grade 1 through grade 2 to grade 3, as reflected in the increasing pairwise distances between grades (Fig. 2c, e). In the IMC - METABRIC dataset, although the distance between grade 1 and grade 3 embeddings was not significantly greater than that between grade 2 and grade 3, the median distance from grade 1 to grade 3 was still higher, suggesting a partial preservation of grade ordering (Fig. 2i). This interpretation is further supported by principal component analysis of the learned embeddings. The first principal component (PC1) revealed a graded separation across tumor grades: grade 3 samples were shifted toward the positive end of PC1, grade 1 clustered toward the negative end, and grade 2 was distributed between them. This suggests that the model captures a latent, continuous trajectory consistent with tumor severity (Fig. 2g). Furthermore, to assess whether the learned embeddings also captured prognostic signals, we examined the association between the first principal component and disease-specific patient survival and indeed found a correlation even within samples of the same tumor grade (Fig. 2d, h). This was reflected in the right-censored concordance index, which yielded median values consistently above 0.5 across cross-validation runs of the selected model (*IMC - Jackson*: 0.55, 0.54, 0.57 for grades 1, 2, and 3; *IMC - Metabric*: 0.86, 0.62, 0.53; Supp. Figure 3a, b). These analyses show that the graph models learned meaningful, even clinically interesting, sample representations that go beyond the separation of labels they were trained on and offer two important implications: (1) the multifaceted utility of these embeddings suggests that the models base their predictions on biologically meaningful features and may generalize to further interpretation tasks, and that (2) the continuous nature of the learned representations reflects gradual variability across tumor grades and patient subgroups, highlighting their potential for future studies to explore clinical outcomes along such latent trajectories.

## Uncovering spatial patterns of immune cell distribution in breast cancer with graph neural networks

These results suggest that gene expression states, the input node states in the presented ablation study, contain significant information about the tissue labels, even in the absence of information about spatial connectivity. However, the cell-wise gene expression states

**Table 2 | Hyperparameters related to model topology for models with molecular features as input screened in grid search for each data set**

| dataset | depth feature embedding | width feature embedding | depth node embedding | width node embedding | depth graph embedding | width graph embedding |
|---|---|---|---|---|---|---|
| IMC - breast cancer (Jackson) | {1, 2, 3} | {4, 8, 16, 32, 64} | {1, 2, 3} | {4, 8, 16, 32, 64} | {1, 2, 3} | {16, 64} |
| IMC - breast cancer (METABRIC) | {1, 2, 3} | {4, 8, 16, 32, 64} | {1, 2, 3} | {4, 8, 16, 32, 64} | {1, 2, 3} | {16, 64} |
| CODEX - colorectal cancer | {1, 2, 3} | {4, 8, 16, 32, 64} | {1, 2, 3} | {4, 8, 16, 32, 64} | {1, 2, 3} | {16, 64} |

*node embedding*: The node embedding describes transformations of node-wise feature vectors and is used in MI and GCN/GIN models. All layers have the same width. *graph embedding*: The graph embedding describes the layer stack that transforms the graph representation to a graph label prediction. The input representation derives from an aggregation over nodes for MI and GCN/GIN models and is the input feature vector for MLP models. All layers have the same width.

themselves are functions of the spatial context[9,17], thus potentially confounding this ablation result. To address this potential limitation in the capture of spatial patterns of cells that are predictive of tissue labels, we set out to perform a similar ablation study in which the input node states are discrete cell type labels. These cell type labels do not resolve fine-grained gene expression variability within cell types that is often confounded by spatial context[9] but may still represent relevant spatial patterns in the tissue: for example, the spatial distribution of immune cells within tumors in a spatial graph of cells that has cell type labels as node states. To specifically query immune cell distributions, we reused the previously described graph representation of tissue images, replacing molecular expression profiles with binary immune versus non-immune cell representations as node features.

To determine whether GNNs pick up tumor phenotype specific spatial patterns of immune infiltration into the tumor, we designed a second ablation study to compare their classification performance against baseline models. Specifically, we compared: (1) the graph tissue representation with binary node features of immune vs non-immune cell ("Spatial Tissue Architecture" model), (2) tissue density structure, using either the graph skeleton or the histogram of node degrees without cell phenotype information ("Density" model), and (3) the cell type fractions (immune vs. non-immune) only ("Cell Type Fractions" model) (Fig. 1c). As an additional control, we trained GNNs on data with randomly permuted node labels ("Permuted Spatial Tissue Architecture" model) to test whether predictions relied on specific immune-tumor spatial arrangements, while keeping the adjacency matrices and cell type fractions fixed (Methods).

Notably, we found that GIN models trained on the spatial immune cell distribution of the IMC - Jackson breast cancer dataset significantly outperformed all other models. This included models trained solely on cell type fractions, tissue density features, and permuted node labels ($\Delta$AUPR = 0.072, $p$ = 0.019 for Cell Type Fractions; $\Delta$AUPR = 0.16, $p$ = 5.47e-6 for Density; $\Delta$AUPR = 0.047, $p$ = 0.041 for Permuted, Fig. 3a). Therefore, we conclude that the GIN model successfully captured distinctive spatial patterns of immune cell invasion associated with different tumor grades. In contrast, for IMC - METABRIC dataset, graph models did not outperform baseline models trained solely on immune cell fractions, although they did outperform both the tissue density-based and permuted graph baselines ($\Delta$AUPR = 0.052 and $p$ = 0.14 for Cell Type Fractions, $\Delta$AUPR = 0.12 and $p$ = 6.69e-6 for Density, $\Delta$AUPR = 0.088 and $p$ = 9.72e-3 for Permuted, Fig. 3b). The generally low performance on this dataset, close to random baseline levels, may explain why modeling complex spatial patterns failed to improve prediction performance. This finding highlights the current limitations imposed by small sample sizes in spatial omics datasets. Together, these findings show that while spatial modeling of immune cell organization can enhance phenotype prediction in certain settings, its effectiveness likely depends on dataset size, signal strength, and the degree to which immune spatial patterns are linked to the target phenotype.

To further explore the ability of GNNs to retrieve spatial patterns, we trained a graph attention network (GAT) on the IMC - METABRIC dataset and analyzed the learned interactions between neighboring cell types in the context of tumor grades (Fig. 3c–e, and Supp. Figure 4). Interpreting the weight matrix of the first graph convolutional layer (Methods), similar to how convolutional filters are visualized in image recognition models, revealed biologically meaningful interactions. Specifically, setting the convolutional filters in context with the learned attention mechanism between cell types (Fig. 3c), we found that the proximity of fibroblasts around tumor cells to be indicative of grade 1 tumors, while the occurrence of macrophages next to tumor cells rather indicated grade 3 tumors (Fig. 3c, d). This observation aligns with the increased presence of macrophages near tumor cells in grade 3 tumors and the higher prevalence of fibroblast-tumor cell interactions in grade 1 tumors[16] (Fig. 3e).

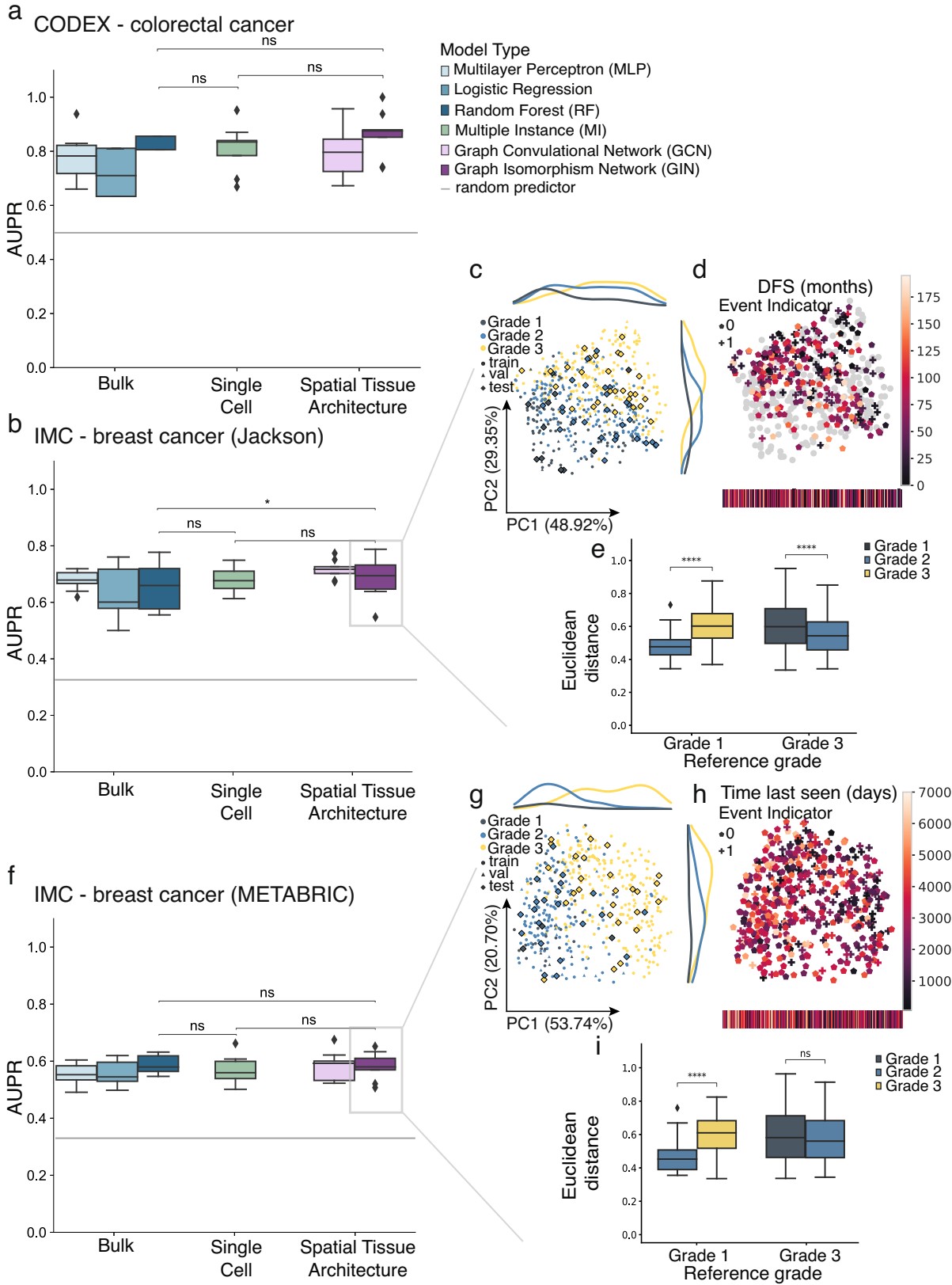

In summary, our findings demonstrate that GNNs applied to immune cell patterns within tumor tissue can identify distinctive spatial patterns of immune infiltration relevant for tumor grade prediction, as shown in the IMC - Jackson dataset. We further demonstrated a way to extract such patterns from fitted models. Our findings emphasize the value of GNNs in integrating diverse factors contributing to phenotype prediction and to uncover subtle spatial patterns that hold promise for advancing our understanding of tumor microenvironments and informing targeted therapeutic strategies.

## Graph neural networks capture complex immune infiltration patterns in colorectal cancer

Previous studies have shown that the spatial distribution of immune cells in colorectal cancer is predictive of disease outcomes and is used

**Fig. 2 | Graph networks capture latent biological signals related to breast cancer grade and patient survival. a, b, f** Multi-modal ablation study on tumor phenotype classification performance using molecular cell representations. Shown is the area under the precision-recall curve (AUPR) across three-fold nested cross-validation for the best performing hyper-parameter set per test split selected based on the train loss for (**a**) *CODEX - colorectal cancer* and the validation loss for (**b**) *IMC - Jackson*, and (**f**) *IMC - METABRIC*. *Bulk* represents the pseudobulk expression per sample, *Single Cell* is the set of molecular expression vectors per tissue image and *Spatial Tissue Architecture* represents the spatial tissue graph representation. The mean positive class prevalence across the 9 cross validation splits is included as a random predictor (grey line). **c, e** and (**g–i**) show analyses on the graph embeddings from the GIN models for *IMC - Jackson* and *IMC - METABRIC*, respectively. **c, g** PCA of the graph embeddings obtained from a GIN model of training, validation, and test data with class labels superimposed. **d, h** Clinical disease-free survival (DFS) annotations. Gray points indicate graphs without recorded survival annotations. **e, i** The average euclidean distances between graph embedding vectors from different classes. Statistical significance was assessed using a two-sided unpaired Student's t-test ($p > 0.05$, ns; $p < 0.05$, *; $p < 0.01$, **; $p < 0.001$, ***). Box plots show the median (center line), the 25th and 75th percentiles (bounds of box), and whiskers extending to the most extreme data points within 1.5× the interquartile range; outliers are shown as individual points.

to stratify tumors[11,14]. This motivated us to investigate whether spatial immune infiltration patterns could again be found to distinguish between tumor cores with tertiary lymphoid structures (TLS) and those with diffuse immune infiltrates (DII).

To model these spatial patterns, we represented tissues again as spatial proximity graphs where nodes were categorized as either immune or non-immune cells. Surprisingly, GNNs trained on this representation did not significantly outperform models trained on permuted node labels, where cell type identities were randomly shuffled ($\Delta$AUPR = 0.004, $p = 0.88$, Fig. 4a). This suggests that immune cell identity contributed little to the model's predictive performance in this setting. Moreover, models trained on the graph structure alone, without any node feature information, outperformed graph models that included either immune status or full molecular profiles as node features ($\Delta$AUPR = 0.023, $p = 0.44$ for Spatial Tissue Architecture (immune/non-immune features); $\Delta$AUPR = 0.027, $p = 0.45$ for Spatial Tissue Architecture (molecular features), Fig. 4a, Supp. Figure 2). This indicates that the underlying spatial arrangement of cells, independent of their molecular or immune identity, may be the dominant predictive signal in this dataset. Supporting this, even random forest classifiers trained on node degree histograms achieved comparable performance in distinguishing tumor areas with tertiary lymphoid structures from those with diffuse immune infiltrates ($\Delta$AUPR = 0.071, $p = 0.12$, Fig. 4a).

The relevance of tissue density in modelling colorectal cancer also explains the strong performance differences between GCN and GIN models in this setting. While GCNs normalize node degrees during message passing, GINs preserve node degree information through sum aggregation. Modifying GCNs to use sum aggregation restored their performance to match GINs ($\Delta$AUPR = 0.17 and $p = 1.49e\text{-}4$ GIN vs. GCN, $\Delta$AUPR = 0.015 and $p = 0.51$ GIN vs. GCN with sum aggregation, Supp. Figure 5d). This finding highlights the significance of preserving tissue density structure in models of cellular organization.

We note that tissue density may be confounded by local cell type composition, thus not guaranteeing that the density model is indeed an ablation that is free of spatial information. To understand if the GIN model captured immune-related spatial features in these settings in which it did not outperform the density model, we employed a gradient-based interpretability approach, calculating the gradient of the model's output with respect to node-level inputs to estimate the contribution of individual cells. Positive gradient values indicated features characteristic of TLS regions, while negative values pointed toward the DII label. First, stratified cells by their node degree and the immune-to-non-immune cell ratio in their local neighborhood and computed average gradient values. Interestingly, while node degree alone was sufficient to achieve high predictive performance, the model clearly incorporated immune identity into its predictions. For instance, cells with node degrees between 10 and 15 and high immune fractions were strong indicators of TLS regions, whereas cells with similar degree but lower immune content were associated with DII regions (Fig. 4b). This level of discrimination could not be achieved by models relying only on tissue density. Next, we asked whether cells within similar local features, same node degree and immune fraction, were

used differently by the model depending on whether they point towards TLS or DII regions. For this, we compared the average gradient values between cells from TLS and DII images (Fig. 4e, f). The gradient values revealed a specific subset of cells (with node degree 10–20, immune cell fraction > 80%) that emerged as key determinants of TLS classification. However, the presence of these cells alone did not fully explain the model's predictions; cells with identical local properties exhibited substantially lower gradient values when they originated from DII images. This suggests that the model leveraged broader spatial context beyond local density and immune abundance. To further investigate potential sources of this contextual difference, we analyzed the neighborhood compositions of these predictive cells, using annotations from the original dataset (Fig. 4c, d, g). We found that cells linked to TLS regions were most often situated in follicle- or T cell-enriched neighborhoods, while the same cell types in DII regions were more frequently embedded in granulocyte- or macrophage-enriched neighborhoods. Even among cells residing in T cell-enriched neighborhoods, gradient values differed markedly between TLS and DII images (Fig. 4h, i), further supporting that the model captures subtle, higher-order spatial cues that distinguish immune microenvironments.

These findings suggest that the graph models capture spatial organization patterns beyond local tissue density, immune cell composition, and immediate neighborhood context. Despite cells having the same node degree, immune fraction, and neighborhood annotation, the model assigns distinct importance depending on whether they originate from TLS or DII regions. This indicates that the model leverages more complex spatial relationships within the tissue subgraph to distinguish between these phenotypes. Such patterns are difficult to capture with conventional models, highlighting the strength of graph-based approaches in learning subtle, context-dependent features of tissue architecture. It is also important to keep in mind that tissue density may still embed some spatial context due to confounding with local cell composition, which may partly account for the performance of the density-based models relative to the graph models.

## Cell type encodings enable interpretability in graph representation learning despite lower prediction accuracy

Cell type labels are a coarsening of vector-shaped cell-wise mean gene expression observations. One would expect the increased feature complexity of gene expression states compared to cell type labels to translate to overall improved predictive ability of models that use these node states in the input. Indeed, graph models were significantly better when trained on gene expression vectors as opposed to one-hot encoded cell type node representations on the breast cancer datasets, *IMC - Jackson* data and *IMC - METABRIC* data ($p = 7.22e\text{-}4$ and $p = 3.59e\text{-}3$, respectively, Supp. Fig. 5a–c). Nonetheless, cell type encodings offer a lower-dimensional and interpretable representation of tissue, which can be particularly valuable for identifying structural patterns and supporting mechanistic studies, such as those modeling immune infiltration.

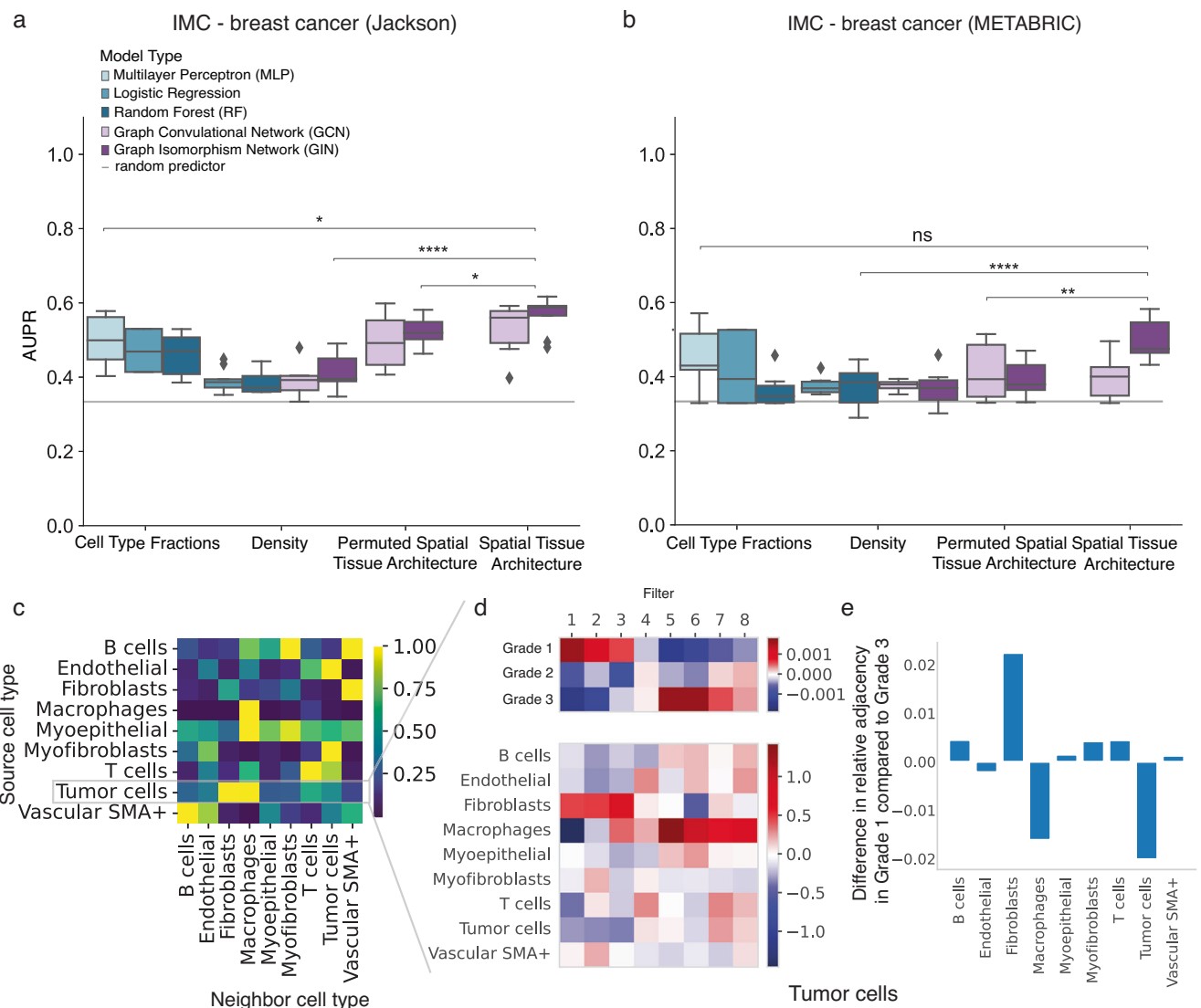

**Fig. 3 | Graph neural networks capture tumor microenvironment features beyond cell type proportions or tissue density. a, b** Multi-modal ablation study on breast cancer tumor grade classification performance using binary cell types feature space for (**a**) *IMC - Jackson*, and (**b**) *IMC - METABRIC*. Shown is the area under the precision-recall curve (AUPR) across three-fold nested cross-validation for the best performing hyper-parameter set per test split selected based on the validation loss. *Cell Type Fractions* represent the ratio between immune vs non-immune cells per sample, *Density* is represented either as the histogram of node degrees within a sample, or it is the full graph structure without node features. *Permuted Tissue Spatial Architecture* refers to a spatial tissue graph representation with cell identities randomly permuted across the graph and *Tissue Spatial Architecture* represents samples via their spatial tissue graphs. The mean positive class prevalence across the 9 cross validation splits is included as a random predictor (grey line). **c–e** Interpretation of the attention mechanism of a GAT model (Methods) trained on

the *IMC - METABRIC* dataset with cell type input. **c** Heatmap of the attention weights between different pairs of key and query cell types. **d, e** Neighborhood analysis on tumor cells. **d** Heatmap of the filter weight matrix of the first convolutional node embedding layer weighted by the attention weights for Tumor query cells (bottom) set into global context by the averaged gradients of the different graph labels with respect to the filter activation scores (top). **e** Difference in neighborhood frequency between tumor cells and other cell types in cancer grade 1 versus grade 3, showing the average differences between the neighboring environment of tumor cells during disease progression. Statistical significance was assessed using a two-sided unpaired Student's t-test ($p > 0.05$, ns; $p < 0.05$, *; $p < 0.01$, **; $p < 0.001$, ***). Box plots show the median (center line), the 25th and 75th percentiles (bounds of box), and whiskers extending to the most extreme data points within 1.5× the interquartile range; outliers are shown as individual points.

## Introducing additional prediction tasks to combat overfitting

Overfitting is of particular concern in relatively small cohorts of hundreds of observations as those that we considered here, especially when working with high-dimensional molecular feature spaces. We introduced an auxiliary self-supervision task (Methods) to the graph model, where the graph neural network predicts the cell type composition of neighboring spectral clusters, to constrain the node embeddings, and compared the resulting performance with standard graph models. However, this task did not improve overall performance of the graph models (Supp. Fig. 6). We added further sample-level labels in a multi-task setup to the GCNs trained on the *IMC - Jackson* and

*IMC - METABRIC* datasets but did not find this to improve the prediction accuracy on test data (Supp. Fig. 7). These results highlight the challenges of training expressive graph models on limited data and point to the need for larger, more diverse datasets to fully leverage their potential.

## Discussion

In this study, we evaluated the ability of GNNs to capture tissue phenotypes from spatial molecular profiling data, leveraging their capacity to implicitly integrate multiple layers of biological information: spatial organization of cells, overall cell type composition, molecular

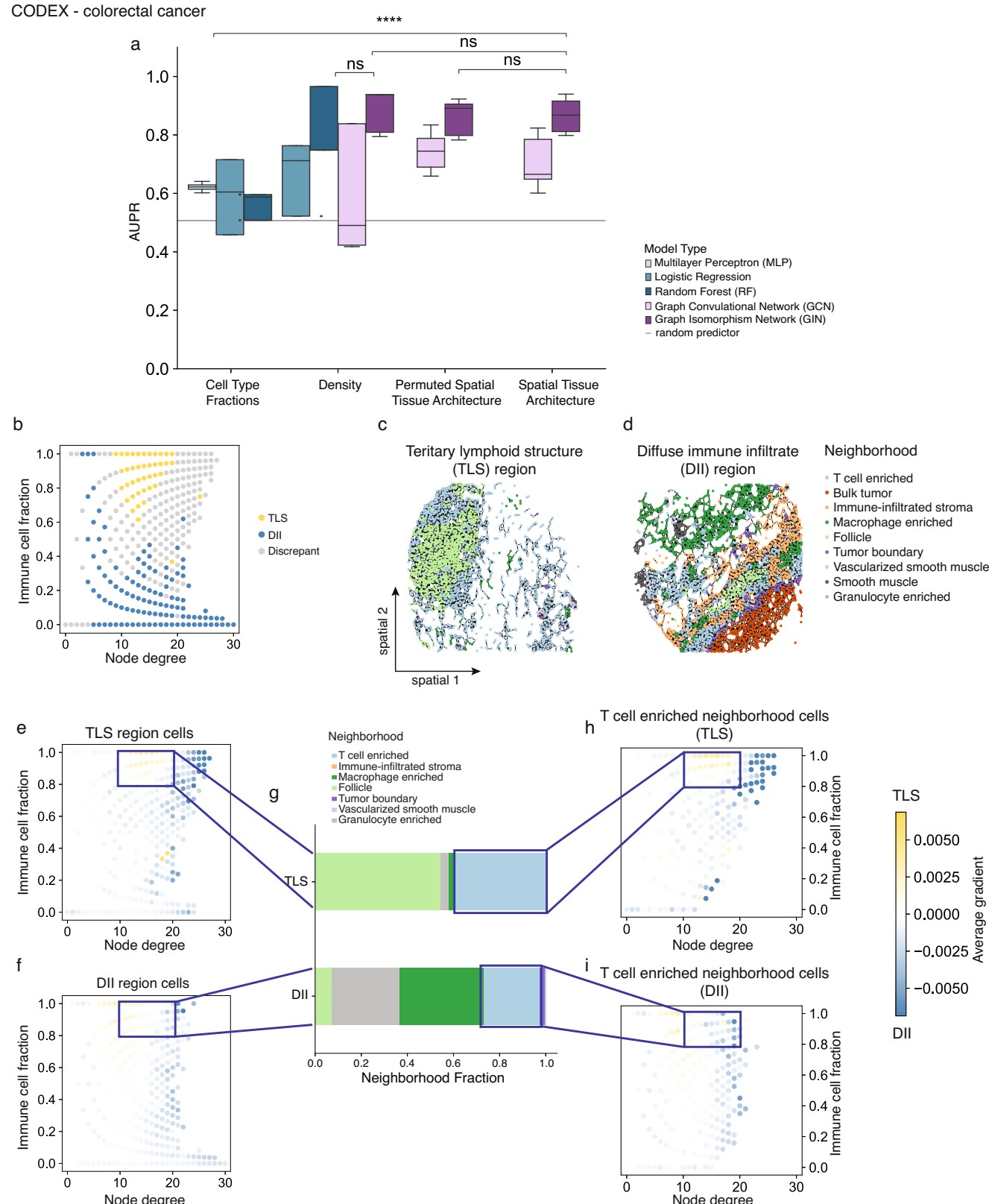

expression profiles as well as structural tissue features such as cellular density patterns. We found that for relatively simple classification tasks, such as predicting tumor grade in breast cancer, the inclusion of spatial context did not improve phenotype classification performance, likely due to the limited dataset sizes of a few hundred images. However, despite this, GNNs successfully captured biologically meaningful spatial patterns, revealing insights beyond what traditional statistical approaches could extract. For instance, GNN-derived sample representations contained clinically relevant signals beyond categorical tumor grade labels. Using Graph Attention Networks, we identified tumor-grade-specific cell type interactions in one of the breast cancer datasets, and in colorectal cancer, GNNs uncovered complex immune infiltration

**Fig. 4 | Graph neural networks model complex spatial immune infiltration patterns in colorectal cancer. a** Multi-modal ablation study on colorectal cancer anatomical phenotype prediction using binary immune vs. non-immune cell feature space. Shown is the area under the precision-recall curve (AUPR) across threefold nested cross-validation for the best performing hyper-parameter set per test split selected based on the train loss. *Cell Type Fractions* represent the ratio between immune vs non-immune cells per sample, *Density* is represented either as the histogram of node degrees within a sample, or it is the full graph structure without node features. *Permuted Spatial Tissue Architecture* refers to a spatial tissue graph representation with cell identities randomly permuted across the graph and *Spatial Tissue Architecture* represents samples via their spatial tissue graphs. The mean positive class prevalence across the 9 cross validation splits is included as a random predictor (grey line). **b** Cells stratified by node degree and fraction of immune cells in the immediate neighborhood colored by agreement between

enrichment of a cell category within one phenotype class and the average model saliency using GIN model. **c, d** Spatial plots of the spatial tissue graphs colored by the neighborhood cell annotation from the original publication from samples with different anatomical phenotypes, **c** TLS region and (**d**) DII regions. **e–i** Cell saliency analysis based on the GIN model. **e, f** Average cell saliencies stratified by node degree and fraction of immune cells in the immediate neighborhood for cells from (**e**) TLS samples and (**f**) DII samples. Saliencies are computed as the gradient of the output with respect to the input cell representations (yellow: TLS, blue: DII). **g** Neighborhood composition of the cells highlighted in (**e**) and (**f**). **h, i** Same as (**e, f**) of cells annotated as T cell enriched neighborhoods. Statistical significance was assessed using a two-sided unpaired Student's t-test ($p > 0.05$, ns; $p < 0.05$, *; $p < 0.01$, **; $p < 0.001$, ***). Box plots show the median (center line), the 25th and 75th percentiles (bounds of box), and whiskers extending to the most extreme data points within 1.5× the interquartile range; outliers are shown as individual points.

patterns that would not have been apparent through simple statistical analyses. Even in cases where these patterns did not enhance classification accuracy, their successful retrieval suggests that GNNs could serve as valuable tools for spatial data analysis, helping to characterize tissue organization principles and microenvironmental interactions.

As datasets grow in size and phenotype complexity, graph-based approaches remain a promising avenue for modeling spatial molecular data. With richer and more extensive datasets, there will be greater flexibility to explore more sophisticated model architectures. While our study was constrained to relatively simple GCN and GIN models due to dataset size limitations, future work could benefit from models with higher capacity, such as Graph Attention Networks[18] or spatially aware message-passing architectures, which may be more sensitive to subtle tissue niche motifs. Additionally, hierarchical pooling strategies could enhance information aggregation in larger graphs[11], which may for example become available in the context of tissue clearing[19]. Integrating spatial profiling with high-resolution single-cell RNA sequencing may further improve the molecular input space for GNNs[20], allowing for a more detailed characterization of tissue organization.

The spatial patterns captured by graph models in this study may hold practical value for both experimental and clinical applications. For instance, grade-associated spatial arrangements or immune infiltration patterns could guide biomarker discovery by highlighting tissue-level features linked to disease progression. These insights may also help design spatial profiling experiments by prioritizing specific tissue regions or compositions for validation. Furthermore, graph-derived embeddings could serve as spatially informed features in multi-modal models for patient stratification or therapy response prediction. More broadly, graph models offer an interpretable framework for studying tissue organization and understanding microenvironmental signals, with potential applications extending beyond classification. Realizing this promise will require larger, more diverse datasets and rigorous validation to ensure robustness and clinical utility. Taken together, our findings highlight both the current limitations and future opportunities of GNNs for spatial omics, positioning them as powerful tools for studying the complex interplay between tissue structure and molecular state[21].

## Methods
### Data
**IMC - Jackson (breast cancer).** The breast cancer dataset (Jackson et al.[15] with 559 images from 350 patients) was measured with IMC. The dataset consists of samples from three breast cancer grades, grade 1 (114 images), grade 2 (214 images) and grade 3 (231 images). Here, 34 proteins in a panel specific to breast cancer microenvironment were simultaneously measured. We used the segmentation provided by Jackson et al. We used the following channels:

1021522Tm169Di EGFR, 1031747Er167Di ECadhe, 112475Gd156Di Estroge, 117792Dy163Di GATA3, 1261726In113Di Histone, 1441101Er168Di Ki67, 174864Nd148Di SMA, 1921755Sm149Di Vimenti, 198883Yb176Di cleaved, 201487Eu151Di cerbB, 207736Tb159Di p53, 234832Lu175Di panCyto, 3111576Nd143Di Cytoker, Nd145Di Twist, 312878Gd158Di Progest, 322787Nd150Di cMyc, 3281668Nd142Di Fibrone, 346876Sm147Di Keratin, 3521227Gd155Di Slug, 361077Dy164Di CD20, 378871Yb172Di vWF, 473968La139Di Histone, 651779Pr141Di Cytoker, 6967Gd160Di CD44, 71790Dy162Di CD45, 77877Nd146Di CD68, 8001752Sm152Di CD3epsi, 92964Er166Di Carboni, 971099Nd144Di Cytoker, 98922Yb174Di Cytoker, phospho Histone, phospho S6, phospho mTOR, Area. Jackon et al. annotated the following cell types: B cells, T and B cells, T cells, macrophages, T cells, macrophages, endothelial, vimentin hi stromal cell, small circular stromal cell, small elongated stromal cell, fibronectin hi stromal cell, large elongated stromal cell, SMA hi vimentin hi stromal cell, hypoxic tumor cell, apoptotic tumor cell, proliferative tumor cell, p53+ EGFR+ tumor cell, basal CK tumor cell, CK7 + CK hi cadherin hi tumor cell, CK7 + CK+ tumor cell, epithelial low tumor cell, CK low HR low tumor cell, CK + HR hi tumor cell, CK + HR+ tumor cell, CK + HR low tumor cell, CK low HR hi p53+ tumor cell and myoepithelial tumor cell. We coarsened the cell types into B cells, T and B cells, T cells, macrophages, T cells, macrophages, endothelial, stromal cells (vimentin hi stromal cell, small circular stromal cell, small elongated stromal cell, fibronectin hi stromal cell, large elongated stromal cell, SMA hi vimentin hi stromal cell) and tumor cells (hypoxic tumor cell, apoptotic tumor cell, proliferative tumor cell, p53+ EGFR+ tumor cell, basal CK tumor cell, CK7 + CK hi cadherin hi tumor cell, CK7 + CK+ tumor cell, epithelial low tumor cell, CK low HR low tumor cell, CK + HR hi tumor cell, CK + HR+ tumor cell, CK + HR low tumor cell, CK low HR hi p53+ tumor cell, myoepithelial tumor cell). We binarized the cell types into immune cells (B cells, T cells, macrophages) and non immune cells (endothelial, vimentin hi stromal cell, small circular stromal cell, small elongated stromal cell, fibronectin hi stromal cell, large elongated stromal cell, SMA hi vimentin hi stromal cell, hypoxic tumor cell, apoptotic tumor cell, proliferative tumor cell, p53+ EGFR+ tumor cell, basal CK tumor cell, CK7 + CK hi cadherin hi tumor cell, CK7 + CK+ tumor cell, epithelial low tumor cell, CK low HR low tumor cell, CK + HR hi tumor cell, CK + HR+ tumor cell, CK + HR low tumor cell, CK low HR hi p53+ tumor cell, myoepithelial tumor cell). We used the disease-free survival annotations censored in the cases where the disease-free survival equaled the overall survival to perform survival analysis.

**IMC - METABRIC (breast cancer).** The breast cancer METABRIC cohort (Ali et al.[16] with 500 images from 467 patients) was collected with IMC. Here, 37 proteins in formalin-fixed, paraffin-embedded breast tumor samples were measured. METABRIC dataset consists of images from three breast cancer grades, grade 1 (50 images), grade 2

(181 images) and grade 3 (269 images). Ali et al. segmented the single cells in the images using random forest classifier and then the expression of proteins in single cells was quantified. The mean protein expression of the segmented cells is used as the node features of the spatial graph. We used the following channels: HH3_total, CK19, CK8_18, Twist, CD68, CK14, SMA, Vimentin, c_Myc, HER2, CD3, HH3_ph, Erk1_2, Slug, ER, PR, p53, CD44, EpCAM, CD45, GATA3, CD20, Beta_catenin, CAIX, E_cadherin, Ki67, EGFR, pS6, Sox9, vWF_CD31, pmTOR, CK7, panCK, c_PARP_c_Casp3, DNA1, DNA2, H3K27me3, CK5, Fibronectin. Ali et al. annotated the following cell types: B cells, Basal CKlow, Endothelial, Fibroblasts, Fibroblasts CD68 + , HER2 + , HR + CK7-, HR + CK7- Ki67 + , HR + CK7- Slug + , HR- CK7 + , HR- CK7-, HR-CKlow CK5 + , HR- Ki67 + , HRlow CKlow, Hypoxia, Macrophages Vim+ CD45low, Macrophages Vim+ Slug + , Macrophages Vim+ Slug-, Myoepithelial, Myofibroblasts and T cells, Vascular SMA + . We coarsened the cell types into B cells, Endothelial, Fibroblasts (Fibroblasts, Fibroblasts CD68 + ), Macrophages (Macrophages Vim+ CD45low, Macrophages Vim+ Slug + , Macrophages Vim+ Slug-), Myoepithelial, Myofibroblasts, T cells, Vascular SMA+ and Tumor cells (HER2 + , HR + CK7-, HR + CK7- Ki67 + , HR + CK7- Slug + , HR- CK7 + , HR- CK7-, HR- CKlow CK5 + , HR- Ki67 + , HRlow CKlow, Hypoxia). We binarized the cells types into immune cells (B cells, Macrophages Vim+ CD45low, Macrophages Vim+ Slug + , Macrophages Vim+ Slug-": "immune cells, T cells) and non-immune cells (Basal CKlow, Endothelial, Fibroblasts, Fibroblasts CD68 + , HER2 + , HR + CK7 + , HR + CK7- Ki67 + , HR + CK7-Slug + , HR- CK7 + , HR- CK7-, HR- CKlow CK5 + , HR- Ki67 + , HRlow CKlow, Hypoxia). We used the disease-specific survival that is the time until the last follow-up or death censored according to the disease specific death indicator to perform survival analysis.

**CODEX - colorectal cancer.** The colorectal cancer dataset (Schürch et al.[14] with 140 images from 35 patients) was measured with CODEX. The dataset consists of two patient groups, one group with Crohn's-like reaction (CLR) represented in 68 images and one group with diffuse inflammatory infiltration (DII) represented in 72 images. Four regions were sampled from each patient: from patients in the CLR groups, two regions containing a tertiary lymphoid structure (TLS) and two diffuse immune infiltrate regions (DII) were sampled per patient, while from patients in the DII group, four diffuse immune infiltrate regions were sampled per patient. The sample-specific anatomic label (with tertiary lymphoid structure or diffuse immune infiltrate, Fig. 2) Patients from the CLR group have higher overall survival than patients classified as DII. Here, 57 proteins specific to the tumor microenvironment were measured. We used the segmentation previously performed by Schürch et al. The molecular abundance per cell segment and the coordinates of the center of each cell were used to construct the spatial graph. We used the following channels: CD44, FOXP3, CD8A, TP53, GATA3, PTPRC, TBX21, CTNNB1, HLA-DR, CD274, MKI67, PTPRC, CD4, CR2, MUC1, TNFRSF8, CD2, VIM, MS4A1, LAG3, ATP1A1, CD5, IDO1, KRT1, ITGAM, NCAM1, ACTA1, BCL2, IL2RA, ITGAX, PDCD1, GZMB, EGFR, VISTA, FUT4, ICOS, SYP, GFAP, CD7, CD247, CHGA, CD163, PTPRC, CD68, PECAM1, PDPN, CD34, CD38, SDC1, HOECHST1:Cyc_1_ch_1, CDX2, COL6A1, CCR4, MMP9, TFRC, B3GAT1, MMP12. Schürch et al. annotated the following cell types: B cells, CD11b + monocytes, CD11b + CD68+ macrophages, CD11c+ DCs, CD163+ macrophages, CD3 + T cells, CD4 + T cells, CD4 + T cells CD45RO + , CD4 + T cells GATA3 + , CD68+ macrophages, CD68+ macrophages GzmB + , CD68 + CD163+ macrophages, CD8 + T cells, NK cells, Tregs, adipocytes, dirt, granulocytes, immune cells, immune cells / vasculature, lymphatics, nerves, plasma cells, smooth muscle, stroma, tumor cells, tumor cells / immune cells and undefined, vasculature. We binarized the cell types into immune cells (B cells, CD11b+ monocytes, CD11b + CD68+ macrophages, CD11c+ DCs, CD163+ macrophages, CD3 + T cells, CD4 + T cells, CD4 + T cells CD45RO + , CD4 + T cells GATA3 + , CD68+ macrophages, CD68+ macrophages GzmB + ,

CD68 + CD163+ macrophages, CD8 + T cells, NK cells, Tregs, granulocytes, immune cells, immune cells / vasculature, lymphatics and tumor cells / immune cells) and non-immune cells (adipocytes, dirt, nerves, plasma cells, smooth muscle, stroma, tumor cells, undefined and vasculature).

**Spatial proximity graphs.** We considered spatial neighborhood graphs built with fixed kernel radii across all images. In all datasets considered here, pixel dimensions are fixed across images so that radii defined in pixels correspond to consistent spatial distances across images. We defined a raw adjacency matrix $A$ for each image with entries $a_{ij}$ based on a radius $r$ of a kernel between the position of two cells $i,j$ in 2D space $z_i$, $z_j$:

$$a_{ij} = 1 \; if \; \left\| Z_i - Z_j \right\|_2 < r \; else \; 0$$

**Spectral clustering.** We applied spectral clustering to the spatial graphs by first constructing a $k$-nearest neighbor (kNN) graph using the spatial coordinates $\{z_i\}_{i=1}^n$ of the cells, with $k = 10$. The kNN graph is undirected, where an edge exists between two nodes $i$ and $j$ if either $i$ is among the $k$ nearest neighbors of $j$ or vice versa.

**Label preparation for the self-supervision task.** For each spectral cluster $C_i$, we define the local self-supervision label $y_i \in \mathbb{R}^d$, where d is the number of cell types, as the normalized cell type frequency vector computed over all cells in the neighboring clusters $N(C_i)$, where the neighborhood is defined using cluster connectivity in the original kNN graph.

### Hold-out definitions

We implemented a nested cross validation. For each study, the datasets were split into training (80%), validation (10%) and test (10%) datasets, except for CODEX - colorectal cancer study which was split into only training and test datasets due to the limited number of samples. The split was performed on the patient domain, ensuring that images from the same patients are grouped together in the split. In the nested cross validation, we used 3 random tests and 3 validation splits. The same splits were used for all the models to ensure fair comparison. We used early stopping on the validation loss for the two breast cancer datasets and fixed number of epochs for colorectal cancer dataset. The best models were selected based on their performance in terms of the lowest validation loss (or training loss in case of CODEX - colorectal cancer). To evaluate the models, we compared based on Area Under Precision-Recall curve (AUPR) (section Evaluation metrics). This metric provides an overall assessment of the models' ability to distinguish between the tissue phenotype and capture the balance between precision and recall. To determine the optimal hyperparameters, we employed a grid search strategy where different combinations of hyperparameters were explored as shown in Tables 1 and 2.

### Evaluation

*Evaluation metrics:* We used the area under the precision-recall curve (AUPR): as a metric for classification performance across all classes considered.

$$AUPR = \sum_i (R_i - R_{i-1})P_i \tag{1}$$

where $R_i \, and \, P_i$ are recall and precision respectively for threshold $i$. The score for multi-class is calculated using macro average.

*Evaluation comparison:* We used a two-sided t-test to assess the statistical significance of performance differences between independent scenarios. For each scenario, we identified the best-performing model class by comparing the mean AUPR across repeated runs. $p$ values below 0.05 were considered statistically significant.

## Models

All neural network models used in this study are feed-forward architectures designed to perform graph-level classification. The models take graph-structured inputs or reduced representations thereof and predict phenotype-level outcomes. Depending on the experiment, models were trained using either a cell type feature space (one-hot-encoded categorical input) or a molecular feature space (continuous gene expression values).

In molecular feature space models, we first embed the input node features into a lower-dimensional latent space using a fully connected multilayer perceptron (MLP) with non-linear activation, $h_i = MLP(x_i)$ where $x_i$ are the raw features of node $i$, and $h_i$ is the resulting node embedding.

For graph models, node embeddings are passed through graph neural network (GNN) layers, including GCN, GIN, or Graph Attention Networks, to propagate information through the graph structure. The final graph representation is obtained by pooling node embeddings via mean; in our experiments, we used mean pooling.

Each model is trained for graph-level supervision (e.g., tumor class prediction) using the appropriate loss function depending on the task type: categorical cross-entropy (CCE) for classification, mean squared error (MSE) for regression, binary cross-entropy with logits (BCE) for proportion outputs, and a custom right-censored MSE loss for survival prediction.

All models share a consistent data structure and training pipeline to ensure comparability. Node and graph features are accessed as standardized batch tensors, and predictions are generated through the shared forward API. Optimization is performed using the Adam optimizer with learning rate scheduling.

In addition, we implemented random forest and logistic regression baselines using scikit-learn, trained on the same aggregated graph-level feature representations used by the MLP models.

**Bulk models.** *Pseudobulk multi-layer perceptron networks (MLP):* We implemented a pseudobulk reference model by aggregating cell-wise feature vectors into a single vector per image. For models using the molecular feature space, we computed the mean of each feature across all cells in the image. In the case of models using the cell type feature space, we computed a compositional representation by normalizing the distribution of one-hot encoded cell types across the image, resulting in a frequency-based encoding per cell type. The aggregated input vector passed through a fully connected neural network as described in Table 2 (graph embedding) to obtain the graph-level prediction $y = f(x)$.

*Pseudobulk random forest (RF) and logistic regression models:* Using the same aggregated input representations described above, we trained scikit-learn random forest and logistic regression classifiers for graph-level prediction. Model performance was monitored using log-loss on a validation set.

**Single-cell models.** *Multi-instance networks (MI):* For the multi-instance reference model, each node's features $x_i \in \mathbb{R}^d$ were independently transformed using a stack of fully connected layers with non-linear activation functions. At each layer $l$, the transformation is given by, $h_i^{(l+1)} = \phi(h_i^{(l)} W^{(l)} + b^{(l)})$, where $\phi$ is a non-linear activation, $W^{(l)} \in \mathbb{R}^{d_l \times d_{l+1}}$ is a learnable weight matrix, and $b^{(l)}$ is a bias term. After the final layer, the node embeddings $h_i^{(L)}$ are aggregated using a pooling operation (mean, max, or sum) to form a graph-level representation, $z = Pool(\{h_i^{(L)}\}_{i=1}^N)$, This graph embedding $z$ was then passed through a multilayer perceptron to generate the graph-level prediction $y = f(z)$, as detailed in Table 2.

*Correlation Network:* We constructed a correlation network by generating k-nearest neighbor (KNN) graphs based on gene expression similarities instead of spatial proximity. We applied a log transformation to the expression matrix and used

Scanpy's sc.pp.neighbors to compute the correlation-based adjacency matrix.

**Spatially-aware models.** *Node degree models (Density):* To explore the impact of spatial information on the tissue-level phenotype classification, we implemented two models: a random forest and logistic regression. To generate the node degree distributions, we computed the histogram of node degrees from 0 to 14, with an additional bin for nodes with a node degree exceeding 14. The resulting histogram was normalized to obtain the proportion of nodes within each bin. These normalized node degree distributions were then used as input features for the random forest and logistic regression models. By incorporating the full distribution of node degrees per graph, we aim to capture the spatial information within each graph which play a significant role in the graph structure and should be able to provide insights about the tissue-level phenotypes.

**Graph models.** *Graph convolutional networks (GCN):* The node embedding layers for the Graph Convolutional Network are defined as: $H^{(l+1)} = \phi(A^* H^{(l)} W^{(l)})$, where $\sigma$ is a Leaky ReLU activation function with negative slope factor 0.1, $H^{(l)} \in \mathbb{R}^{n \times d}$ is the input node feature matrix of dimensions (number of nodes x input features), $W^{(l)} \in \mathbb{R}^{d \times d'}$ is a learnable weight matrix is a weight matrix of dimensions (input features x output features) and $A^*$ is the symmetrically normalized adjacency matrix: $A^* = D^{-\frac{1}{2}} A D^{-\frac{1}{2}}$ where $A$ is the raw adjacency matrix and $D$ is the degree matrix of A. he resulting node embeddings were aggregated using a pooling layer and passed through a multilayer perceptron (MLP) to obtain graph-level predictions. We additionally implemented a GCN variant with sum aggregation (GCN_SUM) to assess its performance relative to other graph models (Supp. Fig. 5d).

*Graph isomorphism networks (GIN):* We used GIN as a graph neural network model for tissue-level classification, designed to capture global graph structures through aggregation-invariant operations and non-linear transformations. The node embedding layers for the GIN models at layer $l$ are defined as:

$$h_i^{(l)} = MLP\left((1 - \epsilon) h_i^{(l-1)} + \sum_{j \in \mathcal{N}(i)} h_j^{(l-1)}\right) \quad (2)$$

where $h_i^{(l-1)}$ denotes the node feature embeddings vector node $i$ at layer $l-1$, $\mathcal{N}(i)$ denotes the set of neighbors of nodes and $\epsilon$ is a learnable scalar (fixed to 0 in our implementation). To form the graph-level representation, node embeddings from all GIN layers are concatenated, $h_i = CONCAT(h_i^{(0)}, h_i^{(1)}, ..., h_i^{(L)})$, where $L$ is the number of GIN layers. These concatenated node embeddings are then aggregated across all nodes using mean pooling $h = \frac{1}{N} \sum_{i=1}^N h_i$. The graph-level representation $h$ then passed through a fully connected network to learn the tissue-level phenotype classification $y = f(h)$, where $y$ is the classification output.

*Graph convolutional networks with self-supervision (GNN-SS):* We introduced an auxiliary self-supervision task to the graph neural networks. For each pre-computed spectral cluster in the graph, the model predicts the cell type composition of all neighboring clusters combined. After the graph embedding layers, node embeddings within each cluster are pooled and passed through a one-layer neural network to produce the predicted composition. A mean squared error (MSE) loss is computed between the predicted and true composition vectors and added to the main graph-level loss during training.

*GNN-Permuted:* We used the described Graph Neural Networks (GNN) but trained them with randomly permuted node features, preserving the adjacency matrix to maintain the graph structure.

*Graph attention networks (GAT):* We implemented Graph attention networks with dot-product attention to allow for actual incorporation of both partners in the attention value computation. Instead of

computing scalar scores for all nodes and taking pairwise sums to weight the information transfer between two nodes, here, the attention score is computed as

$$\alpha_{ij} = softmax_j(x_i^T W_Q W_K^T x_j), \qquad (3)$$

with $W_Q$ and $W_K$ being weight matrices corresponding to linear embeddings for the query and key nodes, respectively, and softmax being computed over all neighbors of a query node after subtracting the maximum value. We used 4-dimensional key and query embeddings for the dot product computation. The message passing step is then performed as

$$x_i^k = \sigma\left( W_V^T \left( \sum_{j \in N_i} \alpha_{ij} \, x_j \right) \right) \qquad (4)$$

with a node feature embedding matrix $W_V$ and $\sigma$ as a Leaky ReLU activation function with negative slope factor 0.1. Node embeddings were then aggregated using a mean pooling layer followed by a simple MLP for final graph-level predictions.

### Other baselines

*Random predictor:* showing the expected value of the random predictor, which is the mean positive class prevalence plotted per class across the cross-validation folds.

### Downstream analyses and model interpretations

**Tissue graph embeddings.** For the sample representation analyses, we computed the graph embeddings for the whole dataset as obtained as activations after the global node pooling step within the respective GNN. These latent graph representations were then quantile normalized to follow feature-wise uniform distributions as implemented in sklearn.preprocessing.quantile_transform, followed by a PCA transformation.

**Survival analysis.** To assess the signal of the disease-free survival covariate in the graph embedding space, we used the loadings of the first principal component as predictor and quantified the performance using the concordance index for right-censored data. This concordance index computes the fraction of comparable data pairs, that is pairs where at least the earlier event occurred, that were predicted in the correct order.

**Graph attention network interpretation.** We visualized the learned attention weights between nodes of different cell types in the case of one-hot encoded cell type identities as node input features as follows: First, we computed $\widetilde{\alpha_{ij}} = x_i^T W_Q W_K^T x_j$ for all combinations of cell types, then subtracted the maximum value per key cell type, and exponentiated these values to mimic the softmax transformation. For visualization purposes, these values were additionally divided by the maximum value per key cell type. Secondly, inspired by interpretation methods from image recognition, we interpreted the learned filter weights of the first convolutional node embedding layer. We retrieved the node embedding weight matrix $W_V$ and scaled it according to the transformed attention scores corresponding to the key cell type of interest. To set the individual filters into context, we computed the average gradients of the model outputs with respect to the activations of the individual filters. We computed the frequencies of the different cell types as neighbors of a cell type of interest per image and averaged those values over the images per cancer grade to validate findings from interpreting the filter weights. Plotted in Fig. 3e and Supp. Figure 4b are the differences of these averaged frequencies between grade 1 and grade 3.

**Cell saliency analysis.** To determine how much individual cells influence the GNNs predictions, we calculated gradients of the model outputs with respect to the cell type indicators of the input nodes. We deemed cells with positive gradients corresponding to a specific tissue phenotype class as indicative for that phenotype.

### Reporting summary

Further information on research design is available in the Nature Portfolio Reporting Summary linked to this article.

## Data availability

We used published datasets provided in the original studies. Jackson et al.[15]: single cell data https://doi.org/10.5281/zenodo.4607374. Ali et al.[16]: single cell data found in Image Data Resource (https://idr.openmicroscopy.org/) under accession code idr0076. Schürch et al.[14]: single cell data https://doi.org/10.17632/mpjzbtfgfr.1, imaging data https://doi.org/10.7937/TCIA.2020.FQN0-0326.

## Code availability

We summarized all models, training, and interpretation mechanisms discussed here in a Python package centered around graph-level supervision on spatial single-cell graphs (available at https://github.com/theislab/tissue). The code is fully developed by the authors and is released under the BSD-3-Clause License. The archived version is available at Zenodo[22].

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

## Acknowledgements
We would like to thank Jana R. Fischer, Jonas Windhager, and Prof. Dr. Bernd Bodenmiller for assistance with the discussed breast cancer datasets and for discussion on the topic of graph convolutional networks and cancer grade prediction. We would like to thank Eeshit Dhaval Vaishnav Prof. Dr. Aviv Regev for discussion on spatial molecular profiling data. We would like to thank Anna Schaar for discussions about the model and the code base and Louis B. Kümmerle for feedback to the manuscript. This work was supported by the German Federal Ministry of Education and Research (BMBF) under Grant No. 01IS18036B and No. 01IS18053A, by the Wellcome Trust Grant 108413/A/15/D and by the Helmholtz Association's Initiative and Networking Fund through Helmholtz AI [grant number: ZT-I-PF-5-01]. D.S.F. acknowledges support from a German Research Foundation (DFG) fellowship through the Graduate School of Quantitative Biosciences Munich (QBM) [GSC 1006 to D.S.F.] and by the Joachim Herz Foundation. This work was supported in part by funding from the Eric and Wendy Schmidt Center at the Broad Institute of MIT and Harvard. S. R. is supported by the Helmholtz Association under the joint research school "Munich School for Data Science"—MUDS.

## Author contributions
M.A., D.S.F., S.R. and F.J.T. conceived the study. M.A., S.R. and D.S.F. implemented the overall software and performed the analyses. M.A., D.S.F., S.R., A.E. and F.J.T. wrote the manuscript.

## Funding

## Competing interests
F.J.T. consults for Immunai Inc., CytoReason Ltd, Cellarity, BioTuring Inc., and Genbio.AI Inc., and has an ownership interest in Dermagnostix GmbH and Cellarity. The remaining Authors declare no competing interests.
