## [Peer Review File · Nature Communications]

Graph neural networks learn emergent tissue properties from spatial molecular profiles

Corresponding Author: Professor Fabian Theis

Version 0:

Reviewer comments:

Reviewer #3

(Remarks to the Author)

Reviewer 1 commented on the manuscript regarding the clarity of label derivation in the colorectal cancer dataset, the necessity and justification for using graph neural networks (GNNs) in spatial omics data analysis compared to simpler methods, and the fairness of model comparisons considering differences in model complexity and parameterization. Overall, the authors have satisfactorily addressed all major points raised by Reviewer 1. They have made substantial improvements to the manuscript, including methodological clarifications, new control analyses, expanded interpretability, and a balanced discussion of both the advantages and limitations of GNNs in the spatial omics context. The revised work is now more transparent, robust, and compelling. The following comments are for the details:

1. Clarity of label assignment in the colorectal cancer dataset. Reviewer 1 requested clear information on how the Crohn's-like reaction (CLR) and diffuse inflammatory infiltration (DII) labels were obtained. The authors have now explicitly described in the revised Methods section that these labels were assigned based on histopathological assessment of H&E-stained sections, as provided by the original dataset authors (Schürch et al.). The manuscript clearly explains that the distinction between CLR and DII is based on the presence of organized lymphoid aggregates (TLS) versus diffuse immune infiltration, and these labels were obtained directly from the dataset creators. This addition improves the transparency and reproducibility of the study.

2. Justification for spatial transcriptomics and GNNs' computational cost. The reviewer questioned whether spatial transcriptomics and GNNs are necessary for the studied classification tasks, especially when simpler approaches might suffice. The authors have addressed this by (1) acknowledging the limited benefit of GNNs for classification performance in breast cancer datasets (due to limited sample size), and (2) emphasizing the unique interpretability and biological insights provided by GNNs, especially in the colorectal cancer dataset. In particular, GNNs could reveal complex tissue organization patterns, such as immune infiltration and spatial microenvironmental structures, that simpler models cannot capture. The revised manuscript, especially in the Results and Discussion sections, now places greater emphasis on the interpretability and hypothesis-generating value of GNNs, not just their predictive performance.

3. Fairness of model comparisons. Reviewer 1 highlighted potential issues with model comparison due to differences in the number of trainable parameters. The authors responded by stating that all models underwent thorough hyperparameter tuning via nested cross-validation, ensuring that each model had a fair opportunity to perform optimally given its input structure. They justify that matching parameter counts across models with fundamentally different input representations is not meaningful and instead focus on optimizing each approach for its respective data type. Additional analyses, such as multi-task learning and self-supervised strategies, were performed and described in the supplement, further demonstrating that overfitting (not under-parameterization) was the primary concern due to limited data size.

4. Expanded baselines and interpretability analyses. In response to Reviewer 1's request for further controls, the authors included new baseline models (e.g., permuted node labels, tissue density-based models), and extended their interpretability analyses (e.g., gradient-based cell saliency, attention mechanisms in GATs). These analyses strengthen the manuscript's claims and provide a more comprehensive benchmarking of GNNs against simpler alternatives.

5. In addition to commenting on the previous reviewer's critique, I further question the potential clinic usage. While the interpretability analyses are well described, the authors could consider adding a short paragraph (either in the Discussion or Conclusion) that more concretely discusses how the discovered spatial patterns or GNN-derived features might be used in

practical biological or clinical settings. For example, how might these insights inform downstream biomarker discovery or the design of spatial profiling experiments?

(Remarks on code availability)

Point-by-point response to the reviewers' comments

Graph neural networks learn emergent tissue properties from spatial molecular profiles

Mayar Ali^{1,4,5,*}, Sabrina Richter^{1,2,*}, Ali Ertürk^{4,7}, David S. Fischer^{1,2,3,+}, Fabian Theis^{1,2,6,+}

¹Institute of Computational Biology, Helmholtz Zentrum München, 85764 Neuherberg, Germany

²TUM School of Life Sciences Weihenstephan, Technical University of Munich, 85354 Freising, Germany

³Eric and Wendy Schmidt Center at the Broad Institute, Cambridge, MA, 02142, USA

⁴Institute for Tissue Engineering and Regenerative Medicine, Helmholtz Zentrum München, 85764 Neuherberg, Germany

⁵Graduate School of Systemic Neurosciences, Ludwig Maximilian University of Munich, 80539 Munich, Germany

⁶Department of Mathematics, Technical University of Munich, 85748 Garching bei München, Germany

⁷Institute for Stroke and Dementia Research, Klinikum der Universität München, Ludwig-Maximilians-Universität LMU, 81377 Munich, Germany

+ correspondence to fabian.theis@helmholtz-munich.de

* These authors contributed equally

In the following, we present our response to the reviewers comments at the previous journal. We give **comments (black)**, **point-by-point answers (blue)** to the questions and in parts **copy parts of the text or specific panels (brown)**, which directly correspond to comments or reference to them.

Reviewer #3 (Remarks to the Author):

Reviewer 1 commented on the manuscript regarding the clarity of label derivation in the colorectal cancer dataset, the necessity and justification for using graph neural networks (GNNs) in spatial omics data analysis compared to simpler methods, and the fairness of model comparisons considering differences in model complexity and parameterization. Overall, the authors have satisfactorily addressed all major points raised by Reviewer 1. They have made substantial improvements to the manuscript, including methodological clarifications, new control analyses, expanded interpretability, and a balanced discussion of both the advantages and limitations of GNNs in the spatial omics context. The revised work is now more transparent, robust, and compelling. The following comments are for the details:

We thank the reviewer for their thoughtful evaluation of the revised manuscript and their recognition of the substantial revisions made in response to the reviewers' comments. We appreciate the acknowledgment of the improvements in the clarity, robustness, and interpretability of our manuscript.

1. Clarity of label assignment in the colorectal cancer dataset. Reviewer 1 requested clear information on how the Crohn's-like reaction (CLR) and diffuse inflammatory infiltration (DII) labels were obtained. The authors have now explicitly described in the revised Methods section that these labels were assigned based on histopathological assessment of H&E-stained sections, as provided by the original dataset authors (Schürch et al.). The manuscript clearly explains that the distinction between CLR and DII is based on the presence of organized lymphoid aggregates (TLS) versus diffuse immune infiltration, and these labels were obtained directly from the dataset creators. This addition improves the transparency and reproducibility of the study.

We would like to thank the reviewer for the positive comment and for recognising our efforts to clarify the label derivation of the colorectal cancer dataset in the revised manuscript.

2. Justification for spatial transcriptomics and GNNs' computational cost. The reviewer questioned whether spatial transcriptomics and GNNs are necessary for the studied classification tasks, especially when simpler approaches might suffice. The authors have addressed this by (1) acknowledging the limited benefit of GNNs for classification performance in breast cancer datasets (due to limited sample size), and (2) emphasizing the unique interpretability and biological insights provided by GNNs, especially in the colorectal cancer dataset. In

particular, GNNs could reveal complex tissue organization patterns, such as immune infiltration and spatial microenvironmental structures, that simpler models cannot capture. The revised manuscript, especially in the Results and Discussion sections, now places greater emphasis on the interpretability and hypothesis-generating value of GNNs, not just their predictive performance.

We would like to thank the reviewer for acknowledging the improvements made in the revised version to justify the use of GNNs, highlight their limitations, and emphasize the value of their interpretability for generating biological insights.

3. Fairness of model comparisons. Reviewer 1 highlighted potential issues with model comparison due to differences in the number of trainable parameters. The authors responded by stating that all models underwent thorough hyperparameter tuning via nested cross-validation, ensuring that each model had a fair opportunity to perform optimally given its input structure. They justify that matching parameter counts across models with fundamentally different input representations is not meaningful and instead focus on optimizing each approach for its respective data type. Additional analyses, such as multi-task learning and self-supervised strategies, were performed and described in the supplement, further demonstrating that overfitting (not under-parameterization) was the primary concern due to limited data size.

We appreciate the reviewer for acknowledging the improvements made in the revised version and the additional analyses performed to address the reviewers' comments.

4. Expanded baselines and interpretability analyses. In response to Reviewer 1's request for further controls, the authors included new baseline models (e.g., permuted node labels, tissue density-based models), and extended their interpretability analyses (e.g., gradient-based cell saliency, attention mechanisms in GATs). These analyses strengthen the manuscript's claims and provide a more comprehensive benchmarking of GNNs against simpler alternatives.

We would like to thank the reviewer for appreciating the improvements made in the revised version and the additional interpretations analyses performed to address the reviewers' comments.

We are pleased that these changes were deemed satisfactory and have contributed to a more transparent and robust manuscript.

5. In addition to commenting on the previous reviewer's critique, I further question the potential clinic usage. While the interpretability analyses are well described, the authors could consider adding a short paragraph (either in the Discussion or Conclusion) that more concretely discusses how the discovered spatial patterns or GNN-derived features might be used in practical biological or clinical settings. For example, how might these insights inform downstream biomarker discovery or the design of spatial profiling experiments?

We thank the reviewer for this valuable suggestion. As recommended, we have extended the Discussion to elaborate on the potential translational impact of GNN-derived spatial patterns.

The spatial patterns captured by graph models in this study may hold practical value for both experimental and clinical applications. For instance, grade-associated spatial arrangements or immune infiltration patterns could guide biomarker discovery by highlighting tissue-level features linked to disease progression. These insights may also help design spatial profiling experiments by prioritizing specific tissue regions or compositions for validation. Furthermore, graph-derived embeddings could serve as spatially informed features in multi-modal models for patient stratification or therapy response prediction. More broadly, graph models offer an interpretable framework for studying tissue organization and understanding microenvironmental signals, with potential applications extending beyond classification. Realizing this promise will require larger, more diverse datasets and rigorous validation to ensure robustness and clinical utility. Taken together, our findings highlight both the current limitations and future opportunities of GNNs for spatial omics, positioning them as powerful tools for studying the complex interplay between tissue structure and molecular state.